# Transparency in peer review: Exploring the content and tone of reviewers' confidential comments to editors

**Bridget C. O'Brien**[1]*, **Anthony R. Artino, Jr**[2], **Joseph A. Costello**[3], **Erik Driessen**[4], **Lauren A. Maggio**[3]

**1** Department of Medicine and Office of Medical Education, University of California San Francisco, San Francisco, California, United States of America, **2** Department of Health, Human Function, and Rehabilitation Sciences, the George Washington University School of Medicine and Health Sciences, Washington, DC, United States of America, **3** Department of Medicine, Uniformed Services University of the Health Sciences in Bethesda, Bethesda, Maryland, United States of America, **4** Department of Educational Development and Research, School of Health Profession Research, Maastricht University, Maastricht, the Netherlands

* bridget.obrien@ucsf.eduj

**Data Availability Statement:** Data available at: https://doi.org/10.5281/zenodo.5128723.

**Funding:** The author(s) received no specific funding for this work.

## Abstract

### Purpose

Recent calls to improve transparency in peer review have prompted examination of many aspects of the peer-review process. Peer-review systems often allow confidential comments to editors that could reduce transparency to authors, yet this option has escaped scrutiny. Our study explores 1) how reviewers use the confidential comments section and 2) alignment between comments to the editor and comments to authors with respect to content and tone.

### Methods

Our dataset included 358 reviews of 168 manuscripts submitted between January 1, 2019 and August 24, 2020 to a health professions education journal with a single blind review process. We first identified reviews containing comments to the editor. Then, for the reviews with comments, we used procedures consistent with conventional and directed qualitative content analysis to develop a coding scheme and code comments for content, tone, and section of the manuscript. For reviews in which the reviewer recommended "reject," we coded for alignment between reviewers' comments to the editor and to authors. We report descriptive statistics.

### Results

49% of reviews contained comments to the editor (n = 176). Most of these comments summarized the reviewers' impression of the article (85%), which included explicit reference to their recommended decision (44%) and suitability for the journal (10%). The majority of comments addressed argument quality (56%) or research design/methods/data (51%). The tone of comments tended to be critical (40%) or constructive (34%). For the 86 reviews recommending "reject," the majority of comments to the editor contained content that also

**Competing interests:** Three authors have editorial roles in Perspectives on Medical Education: Erik Driessen (Editor-in-Chief), Lauren Maggio (Deputy Editor-in-Chief), and Anthony Artino (Associate Editor). Driessen and Maggio receive an honorarium for their editorial roles. This does not alter our adherence to PLOS ONE policies on sharing data and materials.

appeared in comments to the authors (80%); additional content tended to be irrelevant to the manuscript. Tone frequently aligned (91%).

## Conclusion

Findings indicate variability in how reviewers use the confidential comments to editor section in online peer-review systems, though generally the way they use them suggests integrity and transparency to authors.

## Introduction

The peer-review process serves the dual purposes of assisting editors with publication decisions and providing authors with constructive feedback [1–3]. Journals vary in their approach to peer review, particularly in decisions about revealing the identity of reviewers to authors and authors to reviewers [1, 4, 5]. For many journals, masking reviewer identities became the norm as a way to encourage honest critique without fear of retaliation from authors. However, the soundness of this approach has come into question as such masking may inadvertently encourage reviewers to be less conscientious about providing constructive and respectful feedback to authors [6]. To mitigate this concern, many journals provide guidelines for reviewers that set expectations for quality feedback [7, 8]. In addition, to enhance transparency and accountability, some journals have an open peer-review process that either requires or gives reviewers the option to share their identity and/or make their review publicly available [4, 9]. While such efforts to increase transparency are thought to improve review quality by increasing reviewers' sense of accountability, a series of studies conducted by the *BMJ* found no difference in the quality of masked versus unmasked peer reviews [10–12]. The only statistically significant finding was that reviewers were more likely to decline the invitation to review if they knew their identity would be revealed to authors [11, 12]. An important caveat in the transparency hypothesis is that most journals provide space for reviewers to provide confidential comments to the editor, which effectively allows authors to express concerns they feel uncomfortable sharing with authors. Reviewers' decision recommendations are also often not shared with authors. Concerns can arise among authors when confidential comments and recommendations to the editor result in an editor's decision to reject a manuscript and the rationale for the decision is not shared with the authors.

Reviewer guidelines provide variable guidance about how to use the confidential comments to the editor section. The Committee On Publication Ethics (COPE) ethical guidelines for peer reviewers [13] advises reviewers to *"ensure your confidential comments and recommendations for the editor are consistent with your report for the authors; most feedback should be put in the report that the authors will see."* The guidelines further remind reviewers that these comments *"should not be a place for denigration or false accusation, done in the knowledge that the authors will not see your comments"* [13 p.4]. The journal *Academic Medicine* provides a comprehensive guide for reviewers of research manuscripts that describes how reviewers can use that section [14]. The guide suggests using the confidential comments to the editor to:

> *"recommend additional review by someone with specific expertise, make specific comments on the quality of the manuscript, provide opinions about the relevance or significance of the work, or raise potential ethical concerns. The reviewer can also use this space to give the editor more nuanced and detailed information and to explain the severity of any problems detected*

*in the manuscript, along with the likelihood that the authors can address the problems through a revision. The reviewer should ensure that his or her confidential comments to the editor do not contradict the comments directed toward the authors and that they provide information that is relevant only to the editor." [14 p.10]*

By contrast, *PLOS One* explicitly prohibits confidential comments to the editor and instructs authors to only use this section to declare competing interests [15]. *"Confidential concerns relating to publication or research ethics"* are handled outside the peer review system, through email to the journal [15]. Despite these guidelines, we know little about how reviewers use this section, which leaves a crucial gap in understanding transparency and accountability in peer review.

From authors' perspectives, submitting a manuscript for publication is, at minimum, an opportunity to receive feedback to improve their work. However, for reasons described above, this feedback can be confusing or unhelpful when the feedback received does not align with an editor's decision to reject the manuscript or offers little insight into their decision. This lack of transparency and alignment warrants further consideration for two reasons. First, honest and constructive feedback from reviewers plays a key role in improvement of scholarship. Second, non-transparency (whether warranted or not) raises doubts about the legitimacy and fairness of the peer-review process and suggests an area for process improvement.

To better understand issues related to transparency and quality feedback in peer review, we examined how reviewers use the confidential comments section of an online peer-review system. Our specific research questions are:

1. How do reviewers use the confidential comments to the editor section during peer review?

2. To what extent does the content and tone of reviewers' comments to the editor align with their comments to the authors when reviewers make a "reject" recommendation?

## Materials and methods

### Design

We conducted a descriptive study of reviewers' comments on manuscripts submitted to the health professions education journal *Perspective on Medical Education* (PME). We selected to study PME, which is an established medical education journal (founded in 1988), in light of its commitment to the study of its own processes and because our research team had permission to utilize the data necessary to answer the research questions. Additionally, our author team, AA (Associate Editor), ED (Editor-in-Chief), LM (Deputy Editor-in-Chief), all hold editorial roles at PME. Together we have over 20 years of experience running the journal, which affords our author team with knowledge about the journal's processes and history. Our study was approved by the Ethical Review Board of the Netherlands Society for Medical Education, and the need for informed consent of reviewers and authors was waived. File number: NERN 2020.1.5. Date of decision 18-07-2020.

### Data source

PME is a peer-reviewed journal published by Bohn Stafleu Loghum, a subsidiary of Springer Publishing. The journal aims to support and enrich collaborative scholarship between educators and clinicians, and to advance knowledge regarding clinical education practices. PME believes in taking an evidence-based approach to publishing and thus notifies authors and reviewers on its website and in its author instructions that the journal reserves the right to conduct research on all materials submitted to the journal to improve its processes.

PME publishes on average 65 articles per year, which are a combination of publication types, including original research. In 2019, the journal received 402 submissions. The journal's two deputy editors screen submitted manuscripts and reject a substantial number of the publications prior to peer review. An associate editor handles the manuscripts selected to proceed with peer review and requests at least two external reviews. Some manuscripts receive more than two reviews when an editor feels they need more information about the quality and/or importance of the review. The journal uses single-blind peer review such that the external reviewers are aware of the author's identity, but the authors do not know the reviewers' identity. Reviewers must submit a decision of accept, minor revisions, revisions, major revisions, or reject and provide a narrative statement. They also have the option to submit confidential comments to the editors. The reviewers' comments to the editor are visible to the associate editor, deputy editor-in-chief, and editor-in-chief.

### Inclusion criteria

We included reviews of three peer-reviewed article types: Original Research Articles, Reviews, or Show and Tell (descriptions of educational innovations). All included reviews were of manuscripts submitted between Jan 2019 and August 2020. We only included reviews of new submissions; we did not include any manuscripts that were revised and reviewed again.

### Data collection

The data set was downloaded from PME's Editorial Manager system on August 24, 2020 as an Excel spreadsheet. It included: manuscript ID, submission date, reviewer ID, reviewer country of residence, editor final decision (if available), reviewer comments to the editor, and reviewer comments to the authors. We did not download the names or contact information of reviewers or authors, manuscript title, or manuscript abstract to minimize the likelihood that members of the research team would be able to identify authors or reviewers. The de-identified comments are available on Zenodo (https://doi.org/10.5281/zenodo.5128723).

### Data analysis

We conducted a combination of conventional and directed content analysis on reviewer comments [16]. To reflect the purpose and content covered in confidential comments to the editor, we developed a coding scheme based on studies of peer review [6, 17, 18] and guidelines for peer review (e.g., COPE Guidelines [13], *Academic Medicine* review guide[8]). Our primary codes covered content, section of the manuscript, tone, and alignment between the content and tone of comments to the authors and editors when the reviewer recommended "reject." We added sub-categories as needed via conventional content analysis (deriving categories from the data). Content codes included overall verdict, quality of argument, structure and language, design and methods, reference to comments to authors, summary of manuscript, and other. We also coded the section of the manuscript referenced in the comments. We coded the tone of comments as critical, constructive, supportive, or neutral and noted comments in which the reviewer addressed the editor by name or in a way that conveyed a familiar tone.

To evaluate the alignment between reviewers' comments to the editor and comments to authors, we restricted our analysis to reviews in which the reviewer recommended "reject," as we anticipated that these reviews had the greatest potential for discrepancies between what the reviewers wrote to the authors and to the editors. We coded these reviews for alignment of content and tone. Content was considered not to align if the reviewers mentioned concerns about the manuscript to the editor that they did not mention to the authors. We expected that comments to authors would cover more content than comments to editors and did not

consider this a lack of alignment. Tone was based on overall tone. We coded all reviews that did not contain comments to the editor as aligned with comments to the author. This approach allowed us to calculate the overall incidence of alignment in reviews where reviewers recommended rejecting the manuscript.

To facilitate coding, BCO created an initial code book as informed by the above. BCO, LAM, ARA, ED then each independently coded 15 reviews and then met to discuss their coding, reconcile differences and adjust the code book. Using the revised code book, BCO and LAM independently each coded an additional 35 reviews then met to discuss and revise the code book. The remaining reviews were then assigned to BCO, LAM, ARA and JAC such that each review was coded by two authors. BCO and LAM met with all authors to reconcile codes by discussion.

We calculated descriptive statistics for a) key features of the articles in our review (**Table 1**) and b) key features of reviews with and without comments to the editor (**Table 2**), and c) characteristics of reviews with comments to the editor, including: content (**Table 3**), section of the manuscript, and tone (**Table 4**). Our study is descriptive, so we did not conduct statistical tests to compare reviews with and without comments to the editor. Rather, we provide this information to situate our findings in context.

## Results

Our final dataset included 358 reviews of 168 articles submitted and peer reviewed from January 1, 2019 to August 24, 2020. Each article had 1 to 5 reviews. Articles with only one review reflect situations in which the article was still under review and awaiting an additional reviewer when we extracted the data or rare occasions when an associate editor provided a second review (not captured in the database) due to difficulty recruiting reviewers. The majority of articles were original research articles (65%) and most articles had 2 or more reviews (2.1

**Table 1. Descriptive statistics for 168 articles submitted with completed reviews for *Perspectives on Medical Education* from January 1, 2019 to August 24, 2020.**

|  | No. (%) |
| --- | --- |
| **Year** | |
| 2019 | 115 (68.5) |
| 2020 | 53 (31.5) |
| **Article Type** | |
| Original Research Article | 109 (64.9) |
| Show and Tell [a] | 50 (29.8) |
| Review | 9 (5.4) |
| **Number of Reviews** | |
| 1 | 25 (14.8) |
| 2 | 106 (63.1) |
| 3 | 28 (16.7) |
| 4 | 8 (4.8) |
| 5 | 1 (0.6) |
| **Editor Final Decision** | |
| Accept | 56 (33.3) |
| Reject | 88 (52.4) |
| In Process (reviews complete, awaiting editor decision) | 23 (13.7) |
| Withdrawn | 1 (0.6) |

[a] Show and Tell articles report on an educational innovation and provide preliminary data.

**Table 2. Descriptive statistics for 358 reviews of 168 articles submitted to *Perspectives on Medical Education* in 2019 or 2020, by reviews with comments to editor and reviews with no comments to editor.**

| | Comments to Editor (n = 176) | No Comments to Editor (n = 182) |
|---|---|---|
| | No. (%) | No. (%) |
| **Article Type** | | |
| Original Research Article | 106 (60.2) | 120 (65.9) |
| Show and Tell | 64 (36.4) | 50 (27.5) |
| Review | 6 (3.4) | 12 (6.6) |
| **Reviewer Country [a]** | | |
| Australia | 13 (7.4) | 16 (8.8) |
| Brazil | 2 (1.1) | 3 (1.6) |
| Canada | 45 (25.6) | 33 (18.1) |
| Netherlands | 29 (16.5) | 38 (20.9) |
| New Zealand | 4 (2.3) | 14 (7.7) |
| South Africa | 1 (0.6) | 4 (2.2) |
| Sweden | 2 (1.1) | 5 (2.7) |
| United Kingdom | 10 (5.7) | 15 (8.2) |
| United States | 59 (33.5) | 46 (25.3) |
| **Reviewer Recommendation** | | |
| Accept | 7 (4.0) | 11 (6.0) |
| Minor Revision | 25 (14.2) | 45 (24.7) |
| Revision | 42 (23.9) | 30 (16.5) |
| Major Revision | 53 (30.1) | 58 (31.9) |
| Reject | 48 (27.3) | 38 (20.9) |
| Terminated | 1 (0.6) | 0 |
| **Editor Final Decision** | | |
| Accept | 57 (32.4) | 64 (35.2) |
| Reject | 99 (56.3) | 94 (51.6) |
| In Process | 18 (10.2) | 24 (13.2) |
| Withdrawn | 2 (1.1) | 0 |

[a] Includes countries with 5 or more reviews.

reviews, on average). Among the 144 articles with a final decision from the editor, more than half were rejected (88/144, 61%). **Table 1** displays descriptive information on included articles.

## Characteristics of reviews with and without confidential comments to the editor

Our analysis focuses on reviewer comments, so for the remainder of the results the unit of analysis is the number of reviews (n = 358). Slightly less than half of the reviews included confidential comments to the editor (49%). **Table 2** displays descriptive information on reviews with and without comments. Show and Tell manuscripts had a higher percentage of comments to the editor than Original Research and Review manuscripts. Among reviews that had comments to the editor, a smaller percentage recommended minor revisions (14%) and a larger percentage recommended reject (27%) compared to reviews with no comments to the editor (25% minor revisions, 21% reject). The proportion of reviews with and without comments to the editor that recommended accept, revisions, and major revisions were similar. The proportion of reviews in which the editor ultimately rejected the manuscript was similar for reviews with and without comments (56% with comments, 52% without).

**Table 3. Content of comments: Categories with frequency, description, and example comments for each category.**

| Content Category, Frequency, and Description | Example Quotes |
|---|---|
| Overall verdict (n = 149, 85%)<br><br>An overall comment on the submission as a whole, its originality, contribution to knowledge and the acceptability of the claims, and whether it meets the standards for journal publication. | The paper is well-written, logical, and methodically rigorous. The problem and research questions are well delineated, methodology well reasoned, conclusions and limitations appropriate. Overall, the study adds to understanding of (topic). (35-R1) |
| | Overall, this study does not add anything new, novel, or timely. (8-R1) |
| | I'm going to say REJECT, but can imagine a MAJOR REVISIONS if some of the other reviewers and yourself see a silver lining here. (11-R1) |
| Sub-Category of Verdict:<br><br>Explicit Reference to Recommended Decision (n = 78, 44%)<br><br>Mentions Recommendation (Accept, Minor Revisions, Major Revisions, Reject) | Great article- would be perfectly acceptable for publication as is. (73-R2) |
| | Minor revisions need to reflect some changes to grammar, word selection, and expansion of the discussion section. (29-R1) |
| | I think the paper has promise but needs major revisions to strengthen it before publication. (61-R1) |
| | I recommended rejection—the edits seem really substantial and I think the authors would likely need a more substantive evaluation. I could see accepting it with major revisions, though, if they were able to focus the manuscript and evaluation a bit more. (153-R1) |
| Sub-Category of Verdict:<br><br>Suitability for the journal or article type (n = 18, 10%)<br><br>Journal: Does not meet the quality standards of this journal, not appropriate fit for the journal or unsure about fit, recommend a different journal | Suitability for the Journal: My biggest concern is that I don't know how well this will translate to the general readership of the journal. (91-R2) |
| | Even though the topic is interesting and pertinent, the methodology of this study is not sufficiently robust for publication in this journal. (81-R1) |
| | Article Type: This paper would be of interest to the readership if presented as a show and tell. Less so as a research article. (3-R1) |
| Article Type: Not a good fit for the article type, suggest a different type | This paper is not (in my opinion) sufficiently grounded or rigorous for a research paper but rather could be considered as some type of an innovation report. (47-R1) |
| Quality of arguments (n = 98, 56%)<br><br>Whether the submission is persuasive, coherent and lucid for disciplinary readers. Includes reference to appropriate literature, use of conceptual framework, rationale for the study, discussion and interpretation of findings, implications of findings, and comments about providing sufficient details about methods. | Well written article, needs some work before acceptance. Main concerns: 1. Theoretical framework is mentioned but not clear how it was used. 2. "Qualitative research" has to be specified. (26-R2) |
| | To be frank, I found it difficult to follow the author's reasoning in the paper and did my best to piece together the argument in the most charitable way possible. (106-R2) |
| | There's a lot that I think needs tidying up so that the concepts in the paper are clear to the reader. The attempt to link the method of this paper to theory hasn't been done well, and I'm unsure how much they can build on this given the manipulation that was used in this study. Whilst the paper states uniqueness, many of the papers they cite have done a better job of testing the impact of context on [Topic], and their claims about uniqueness appear to be overstated. (90-R1) |
| Research Design / Methods / Data (n = 90, 51%)<br><br>The clarity of the research questions, the nature of the data, how the research was conducted and appropriacy of the analysis; Also quality of data and whether data are appropriate for the research question / constructs, general comments on results | My key concern, as indicated to the authors, is that they state that it is a phenomenological study and yet don't seem to follow these procedures. (109-R1) |
| | I do wonder whether the statistical analysis has been overly complicated to try to achieve the outcomes that the authors may hope to achieve from this questionnaire. (12-R1) |
| | Overall, I do wonder whether the epistemological stance of the research team is mis-aligned and whether a mixed-methods approach may have helped to address my concerns related to the methodological robustness. (12-R1) |
| | Only concerns was that they altered validated questionnaires and although likely passes face validity, they did not discuss this thoroughly in the paper. Would like them to include all scales and to make results more accessible for non-stat savvy readers. (33-R3) |
| Reference to Comments to authors (n = 44, 25%)<br><br>Explicit reference to comments to authors | Hopefully, I have offered a constructive review that will encourage the authors to thoughtfully consider or justifiably refute my suggestions. (17-R2) |
| | In my comments to the authors, I have noted areas where the manuscript can be strengthened, namely with respect to enhancing detail/clarity and rationale around the methodological approach and sampling. (45-R1) |

(*Continued*)

**Table 3.** (Continued)

| Content Category, Frequency, and Description | Example Quotes |
|---|---|
| Structure and language (n = 35, 20%)<br><br>The overall structure of the submission, the length, the adherence to academic conventions and use of grammatical and appropriate language. | I found the manuscript well written, save for some sentence structure and grammatical errors that I imagine the proof editing process might smooth out. (17-R2) |
| | This is generally well written and I would advise acceptance with minor revisions, but the length seems excessive and could be trimmed down some, especially with respect to the introduction as described above. (45-R2) |
| | I'm concerned that the authors didn't follow the instructions for their chosen article type: Show and tell—200 word abstract (they are more than that), <15 references (they are at 17), 1 figure (they use 2) and 1500 word max (they use >1600 words) (39-R1) |
| Summary of the research (n = 11, 6%)<br><br>Reviewer includes a brief summary of the manuscript/research (often seen as an introduction to the review) | *To protect reviewer confidentiality, we have not provided quotes for this code/subcode* |
| Author competence<br>(n = 9, 5%)<br><br>Comments on the abilities of the authors to conduct and present research suitable for publication. | These researchers did a huge job, but evidently without the right supervision. I regret it that they put so much energy and time in this study (2-R1) |
| | We are concerned that the authors' lack of rigor in preparing and editing this manuscript may reflect a potential lack of rigor in the underlying research work. (47-R2) |
| | I am not sure if the author would be able to turn the article around to a publishable level, but I am giving the author the benefit of the doubt. (106-R1) |
| Ethics / Conflict of Interest (n = 5, 3%)<br><br>Concerns about the ethics of the study; Concerns about possible conflicts of interest for authors; Concerns about reviewer conflict of interest | The content summarized from cited papers is quite divergent. This is a serious ethical concern in terms of misrepresentation of sources, and citing merely for the sake of including a citation (52-R2) |
| | According to my opinion this manuscript is not suitable for publication in PME as I see major flaws in the methods, and also a lack of necessary ethical considerations. (143-R1) |
| | [Name] is a co-author of this paper. [Name] and I completed concurrent post-doctoral fellowships together at [Name of University]. I am bringing this relationship to your attention; although I do not believe it puts me in conflict of interest. (162-R1) |
| Other (n = 70, 40%)<br><br>Content that does not fit any of the categories above, organized into three sub-categories. | The Reviewer: |
| | I'm not familiar with some of the stats methods used in this paper—[name of technique] so can't really comment on that aspect of the paper. Otherwise I quite like it! (56-R2) |
| a) Comments about the reviewer<br>b) Comments reflecting the reviewer's engagement<br>c) Comments on the journal's editorial process | I completed this with the help of a junior colleague [Name]. He is learning to peer review with me. (115-R2) |
| | Hopefully you do not find me too strict. I am curious what you as the editor and also the other reviewers think of it. I am willing to send the manuscript with my comments to the authors, if that would be of any help to them. (143-R1) |
| | Engagement: |
| | Thank you very much for the opportunity to serve as reviewer. (85-R1) |
| | I would be willing to review a re-write of this manuscript.(38-R2) |
| | Always welcome for feedback on how to make my reviews better. (15-R1) |
| | Journal's Editorial Process: |
| | It is a bit jarring to see the name of the school redacted in the main text (with the aim of ensuring anonymity) i.e. [INSTITUTION BLINDED FOR REVIEW] when the abstract to the manuscript makes the origin of the project patently clear. (40-R1) |
| | I would like to reiterate the need for an adequate number of words to do justice to the richness of qualitative data, which requires illustration and description (48-R1) |
| | I would like some guide to the 1–100 score so I am more helpful to you. (52-R1) |

**Table 4. Tone of comments: Categories with frequency, description, and example comments for each category.**

| Tone Category, Frequency and Description | Example Quotes |
|---|---|
| **Critical** (n = 71, 40%)<br><br>Critique mostly focuses on problems without offering actionable suggestions. May be harsh or condescending | I recommend that this manuscript be rejected. The methodology, conceptual framework and data analysis are fundamentally flawed. (31-R1) |
| | I think this article has some interesting points to make, but it is a very challenging read (and quite uninspiring) for the first half. I would have stopped reading after page 2. It reads as being written in a quantitative style, while making some nuanced qualitative assertions, which is a bit jarring. (88-R2) |
| **Constructive** (n = 60, 34%)<br><br>Identifies problems along with potential fixes; offers suggestions for improvement; identifies actionable items the authors can address to improve the manuscript | The treatment of sensitizing concepts is minimal in this submission and needs to be more explicit and developed in order to assess the [Methodology] as rigorous. I think this is likely achievable in a revision. (150-R1) |
| | A few findings overstated in terms of language where I think they need to be more tentatively offered due to the limited scope and numbers involved in this study (and that participants were not overtly asked to comment on some of the inferences made). Otherwise I think it is an important contribution that could be useful to many educators. (164-R2) |
| | I think the paper has promise but needs major revisions to strengthen it before publication. I would suggest broadening the statistical analysis and explaining methods, terminology and results. I think this paper potentially adds to the conversation about [topic]. (61-R1) |
| **Supportive** (n = 27, 15%)<br><br>Positive comments about the work, outright praise with nothing to improve the work | I would recommend this with a few amendments as suggested above. It is innovative and interesting—and I think would be very well-cited by other researchers in this area. (104-R1) |
| | Though the sample size is small and geographically homogenous and the word count is approx. 800 words more than permitted 3500, I recommend accepting this manuscript for publication because overall quality of this study is good and the concept—(name of concept) is an emerging concept. There are very few studies on this topic, more research is needed in this area. This study is definitely a step forward in that direction. (64-R1) |
| **Neutral** (n = 11, 6%)<br><br>Summarizes the work OR summarizes comments to the authors OR offers / implies no judgment of the work opinion | I've given my views here as a programme director and I hope they're fair. (33-R2) |
| | Expert statistical review is needed for this paper I believe, before making any decision to publish. (42-R1) |

Reviews included in our study came from 233 unique reviewers from 20 countries. Reviewers completed 1.5 reviews on average within the timeframe of our study (range 1–9 reviews) and 75 reviewers completed 2 or more reviews. 136 unique reviewers provided comments to the editor, though 29 of these reviewers also provided reviews that did not include comments to the editor. A higher proportion of reviews with comments to the editor came from Canada and the USA compared to reviews without comments to the editor.

## RQ1. How reviewers use the "comments to the editor" section of online peer-review platforms

To answer our first research question, we analyzed the content of the 176 reviews containing comments to the editor (49% of all reviews). Reviewers' comments to the editor were much briefer, on average, than their comments to authors (M = 77 words, SD = 87 and M = 433 words, SD = 199, respectively).

We organized our analysis around 3 dimensions of the comments: Content, Section of the Manuscript, and Tone. **Tables 3** and **4** display the frequency of categories under Content and Tone and provide example comments that reflect common ideas in reviewers' comments.

**Content (Table 3).** Less than half the comments to the editor explicitly stated the reviewers' recommendation (44%), though most provided overall evaluative statements that provided a rationale for the recommendation. More than half the comments remarked on the quality of the authors' argument (56%), which included reference to relevant literature, application of a conceptual framework, or lack thereof. Roughly half the comments addressed research design, methods, and data (51%) which included comments on the need for more information or clarification of methods. A small number of comments (10%) raised concerns about suitability for the journal based on quality standards and readership or concerns about fit with the type of article. Few comments mentioned ethical concerns related to the study or raised potential conflicts of interest for the reviewer or authors (3%). Many comments (40%) included content not covered in our initial coding framework. We organized this content into three categories: comments about a) the reviewer (concern about own limitations as a reviewer, being too harsh, being new to reviewing, and performing a group review or review with a mentee), b) the journal's editorial process (feedback, suggestions, and questions related to the journal and the peer review process), c) engagement with the journal (gratitude for the opportunity to review, willingness to review a revision, or offer to write an accompanying commentary).

**Section of the manuscript.** Nearly half the comments (49%) were too general to tell what part of the manuscript was mentioned. When the section could be determined, Methods (37%) and Results (25%) were most common, followed by Discussion (16%) and Introduction (12%). Comments rarely mentioned title (1%), abstract (2%), and manuscript exhibits (i.e., tables, figures, supplemental materials) (5%).

**Tone (Table 4).** The most common tone of comments to the editor was critical (40%). However, tone varied by reviewer recommendation—among comments with a critical tone, nearly half coincided with a recommendation to reject (48%) and nearly one-third with major revisions (32%). Comments with a constructive tone were also frequent (34%), particularly when reviewers recommended revisions (33%), major revisions (30%), or minor revisions (18%). The remaining comments had a supportive (15%) or neutral (11%) tone. A few comments (6%) suggested the reviewer knew the editor (e.g. "Hi [associate editor name], Always welcome feedback on how to make my reviews better. . .Happy to discuss further as needed. [reviewer name]").

## RQ2. Alignment between the content and tone of reviewers' comments to the authors and comments to the editor when reviewers recommend reject

Among the reviews in which the reviewer recommended "reject" (n = 86), 48 had comments to the editor (56%). We considered reviews with no comments to the editor as aligned in both content and tone. For content, we found that more comments aligned with the comments to the authors than did not (80% and 20%, respectively). Among the 17 reviews with content discrepancies, we found comments to the editor that could reduce transparency to the authors. These comments included overall verdict statements such as "We find this manuscript not suitable for publication" (54-R2) which contrasted comments to the authors that suggested potential changes with statements like "We suggest to. . ." and "This can be overcome by. . ." as well as specific concerns about lack of value (e.g. "I don't find the findings meaningful"(13-R1) and "I fear that this study and these outcomes are of little interest to the readers of this journal."(23-R1)) and inadequacy of data (e.g. "I think that the idea of this paper could be salvaged with additional data gathering and reporting as reflected above, which might suggest a "major

revisions" recommendation, but the current data don't seem adequate for publication"(41-R1)). We also found comments to the editor that provided context or commentary on the review (e.g. "I hope this review is not too harsh and if so please let me know and we can adjust." (24-R1) and "I've given my views here as a programme director and I hope they're fair." (33-R2), as well as comments that did not pertain to the manuscript (e.g. questions about the manuscript rating system or whether institution name should be blinded).

The tone of reviewers' comments to the editor and the author generally aligned (91%). We found that the tone in the7 reviews with discrepant tones were more critical in comments to the editor than to the author.

## Discussion

Our study sought to better understand issues related to transparency in peer review by examining how reviewers use the confidential comments to editors. Our analysis shows that roughly half of reviews contained no additional comments to the editor. Among those that did, most were brief and summarized the reviewers' overall evaluation of the manuscript, often referencing their comments to the authors. Comments most often addressed the quality of the arguments and aspects of the research design, methods, or data. The vast majority of comments to the editor had a critical or constructive tone, which roughly aligns with the decision terms recommended by reviewers. In reviews that recommended rejecting the manuscript, the content aligned more often than not and, in many cases, the additional content was not related to the quality of the manuscript. Tone also aligned in most of these comments.

Overall our findings suggest that most reviewers are transparent in their reviews, sharing their feedback with authors even when they judge the manuscript unsuitable for publication and rarely making additional comments to the editor that raise concerns not shared with the authors. These findings add an important piece of information to the current literature on innovations in peer review. A number of journals have shifted to processes that unmask reviewers' identity as either an optional or required part of peer review [19]. Studies examining reviewers' preferences for sharing their identity with authors [20–22] and the effect of such signed reviews on the quality, tone, and time spent on reviews have shown mixed effects [10, 11, 22–24], perhaps reflecting different norms across different fields and reviewer pools. None of these studies indicate whether these unmasked reviewers can write confidential comments to editors. This option provides a mechanism for reviewers to raise concerns regarding ethical issues that warrant further investigation by editors (e.g., plagiarism, duplicate publication) via a confidential channel that likely feels more comfortable to reviewers. Yet this option means that a piece of the peer-review process remains invisible to authors, which could fuel suspicion about bias in the review process. Findings from our exploratory study suggest such suspicions may be unwarranted. Some journals aim to alleviate such suspicions by instructing reviewers to use the comments to editors for a limited set of purposes such as concerns about conflict of interest or unethical breeches [15]. We found very few instances of comments for these purposes, suggesting that such guidance would likely result in infrequent use of confidential comments to editors.

The journal we selected provided minimal guidance to reviewers for how to use the confidential comments to editors, which gives us insight into reviewers' interpretations and potential norms in the field about how one might use this section. That said, the diversity of comments included in this section, and the fact that 51% of reviewers did not use the confidential comments area, suggests that journals might benefit from giving clearer guidance or training to their reviewer pool. Many journals provide guidance and checklists to reviewers, the contents of which were catalogued in a recent scoping review of biomedical journal peer review guides [7]. Unfortunately, this review did not report the number of guides that included

content about how to use the confidential comments section, but may be worth exploring to understand the norms around use of this section and to help standardize language across peer review guides.

Our findings revealed some potential benefits of confidential comments to editors that warrant consideration. Lee and colleagues characterize peer review as an inherently social and partial process—one that cannot and arguably should not attempt to be objective [25]. Our findings align with this characterization and reveal ways that reviewers use these comments to signal engagement in the scholarly community and reflect on their positionality. We found reviewers using this section to express gratitude for the opportunity to review the manuscript and willingness to review a revision. Editors may find value in these expressions of goodwill and commitment to the community. Reviewers also showed signs of self-awareness and desire for improvement by noting concern about the tone of their review and indicating a desire for feedback. Such comments reveal reviewers' thoughtfulness in crafting reviews and may suggest a way to reward and sustain their efforts by giving them ways to hone their skills and learn from editors. Reviewers also occasionally used the confidential comments section to alert editors to their perspective on a topic or limitations regarding evaluation of the content or methodology. Such information may provide helpful context to editors, particularly when needing to make a decision based on disparate reviews. Eliminating or restricting use of this section without providing alternative means and guidance about how best to communicate such information to editors might reduce opportunities for reviewers to communicate with editors in ways that enrich relationships in what can otherwise feel like an impersonal process.

Several articles have described various types of bias and partiality that may apply to peer review, including bias for or against the author based on personal characteristics (race, ethnicity, gender, status) or emotions toward the author (e.g. sympathy for their situation), the status or prestige of the author's institution, and the ideology, philosophical orientation, or methodology put forth by the author [25–27]. We did not evaluate comments specifically for such biases and partiality, though our coding of "other" content could have captured some of this content. Since authors and institutions were deidentified in our dataset, our ability to evaluate this content was limited. That said, this type of analysis could be a fruitful direction for future research, particularly to understand when reviewer partiality is problematic and how comments to the editor may help or hinder awareness of such partiality.

Our findings should be considered in light of several study limitations. First, we analyzed comments from a single journal within the field of medical education. It is possible that in another medical education journal, or one outside of the field, comments to the editor differ from those analyzed here, which limits the generalizability of our findings. Second, we focused on determining alignment between comments to the author and editor for those manuscripts that reviewers recommended rejecting. Future research should consider exploring alignment for those manuscripts recommended to accept and revise. Despite the fact that the majority of comments were written by authors located in English-speaking countries, this study also included comments from reviewers for which English was likely not their first language. This point could have introduced subtle differences in the comments that our predominantly North American research team did not detect. Therefore, we encourage investigators to consider broadening their research team in future work, or engaging with stakeholders from the reviewers' countries to obtain feedback on the findings.

## Conclusion

Findings from our exploratory study indicate that nearly half of reviews contain no confidential comments to the editor and those that do use them in a variety of ways. Most comments

included a summary or rationale for the reviewer's overall evaluation of the manuscript and carried a critical or constructive tone. Reviewers' commitment to transparency, based on lack of use of confidential comments and alignment between the content and tone of reviewers' comments to editors and authors, appears to be strong and in many cases lack of alignment in content reflected concerns unrelated to the manuscript or circumstances the reviewer felt important call to the editors' attention. These findings suggest opportunities for further investigation and discussion about the pros and cons of retaining or clarifying the purpose of confidential comments to the editors as part of peer review.

## Acknowledgments

**Disclaimer:** The views expressed in this article are those of the authors and do not necessarily reflect the official policy or position of the Uniformed Services University of the Health Sciences, the Department of Defense, or the U.S. Government.

## Author Contributions

**Conceptualization:** Bridget C. O'Brien, Anthony R. Artino, Jr, Erik Driessen, Lauren A. Maggio.

**Data curation:** Bridget C. O'Brien, Joseph A. Costello, Lauren A. Maggio.

**Formal analysis:** Bridget C. O'Brien, Anthony R. Artino, Jr, Joseph A. Costello, Erik Driessen, Lauren A. Maggio.

**Investigation:** Bridget C. O'Brien.

**Methodology:** Bridget C. O'Brien, Anthony R. Artino, Jr, Erik Driessen, Lauren A. Maggio.

**Project administration:** Bridget C. O'Brien, Erik Driessen, Lauren A. Maggio.

**Writing – original draft:** Bridget C. O'Brien, Lauren A. Maggio.

**Writing – review & editing:** Bridget C. O'Brien, Anthony R. Artino, Jr, Joseph A. Costello, Erik Driessen, Lauren A. Maggio.

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
