## [Decision Letter · Decision Letter 0]

21 Sep 2021

PONE-D-21-24629Transparency in peer review: Exploring the content and tone of reviewers’ confidential comments to editorsPLOS ONE

Dear Dr. O'Brien,

Thank you for submitting your manuscript to PLOS ONE. After careful consideration, we feel that it has merit but does not fully meet PLOS ONE’s publication criteria as it currently stands. Therefore, we invite you to submit a revised version of the manuscript that addresses the points raised during the review process. Both referee agreed by suggesting that the paper needs minor revisions. From my part, I think that the data associated to your article do not permit a complete replication of your analysis. You should therefore add information to the data file available on Zenodo: 1) the classification of each article used for building Table 3 and Table 4  of the articles; 2)  the recomendation of each referee and the final decision of the editor.

We look forward to receiving your revised manuscript.

Kind regards,

Alberto Baccini, Ph.D.

Academic Editor

PLOS ONE

Journal Requirements:

2. Please provide additional details regarding participant (author and reviewer) consent. In the ethics statement in the Methods and online submission information, please ensure that you have specified whether the need for consent to use reviews for research was specifically waived by the ethics committee.

“I have read the journal's policy and the authors of this manuscript have the following competing interests: Erik Driessen (Editor-in-Chief), Lauren Maggio (Deputy Editor-in-Chief), and Anthony Artino (Associate Editor). Driessen and Maggio receive an honorarium for their editorial roles.”

Reviewers' comments:

Reviewer's Responses to Questions

**Comments to the Author**

1. Is the manuscript technically sound, and do the data support the conclusions?

Reviewer #1: Yes

Reviewer #2: Yes

2. Has the statistical analysis been performed appropriately and rigorously? 

Reviewer #1: N/A

Reviewer #2: Yes

3. Have the authors made all data underlying the findings in their manuscript fully available?

Reviewer #1: Yes

Reviewer #2: Yes

4. Is the manuscript presented in an intelligible fashion and written in standard English?

Reviewer #1: Yes

Reviewer #2: Yes

5. Review Comments to the Author

Reviewer #1: The authors provide a first glimpse into the relationship between reviewer comments to editors and comments to authors. This type of investigation is long overdue and this work provides a first small step in this direction. Having anecdotally experienced myself that reviewer comments to the editor may diverge substantially from the comments to the authors, I have wondered how common this really is. While this is just an analysis of one journal, it is a start and it is reassuring that such divergence may not be all that common. I find the article well written and easy to follow. Methods are explained well and in sufficient detail. I have only a few minor comments:

- L186: "We coded the tone of comments as critical, constructive, supportive, or neutral"

There were no vicious or antagonistic, etc. comments to the authors? Maybe, if this is a very collegial field, one needs to look at journals in more competitive fields to find differences in tone between comments to authors and comments to the editor? Should this surprising (to me) finding not be mentioned?

- L216: "Each article had 1 to 5 reviews." - further up it is mentioned that the journal policy requires two reviews?

- Perhaps it is worth explicitly mentioning the overall/total incidence of diverging editor comments? Reader can calculate it, of course, from the tables, but assuming that no editor comments would mean alignment with author comments, I think it would be worth to explicitly mention the incidence of divergence in the text and maybe even in the abstract: after all, the main message of the work is that it may not be very common, so why not mention the incidence in the abstract?

Reviewer #2: The paper is well-written. The research questions are clear described and answered and conclusions are appropriate. The authors see the limits of this study (analysis of comments of a single journal in a specific field) and suggests possible developments and further investigations.

The authors fail to mention that the journal is open access. This is not a secondary fact with respect to the care of the transparency of the processes and provides an important key to the interpretation of the results. I suggest to add a paragraph on this and to add this among the "limits" of the study (the results would be the same in a closed access journal?)

The dataset is available in Zenodo. Perhaps it would be better a csv and not an xls, according to the FAIR principles.

6. PLOS authors have the option to publish the peer review history of their article (what does this mean?). If published, this will include your full peer review and any attached files.

Reviewer #1: **Yes: **Björn Brembs

Reviewer #2: **Yes: **Paola Galimberti

---

## [Author Response · Author response to Decision Letter 0]

21 Oct 2021

Please see attached document "Authors' Response to Editors and Reviewers"

---

## [Editor Report · Decision Letter 1]

12 Nov 2021

Transparency in peer review: Exploring the content and tone of reviewers’ confidential comments to editors

PONE-D-21-24629R1

Dear Dr. O'Brien,

We’re pleased to inform you that your manuscript has been judged scientifically suitable for publication and will be formally accepted for publication once it meets all outstanding technical requirements.

Kind regards,

Alberto Baccini, Ph.D.

Academic Editor

PLOS ONE
---

## [Editor Report · Acceptance letter]

17 Nov 2021

PONE-D-21-24629R1 

Transparency in peer review: Exploring the content and tone of reviewers’ confidential comments to editors 

Dear Dr. O'Brien:

I'm pleased to inform you that your manuscript has been deemed suitable for publication in PLOS ONE. Congratulations! Your manuscript is now with our production department. 

Kind regards, 

on behalf of

Prof. Alberto Baccini 

Academic Editor

PLOS ONE